# Effect of Air Exposure and Re-Submersion on the Histological Structure, Antioxidant Response, and Gene Expression of *Procambarus Clarkii*

**DOI:** 10.3390/ani13030462

**Published:** 2023-01-28

**Authors:** Xiangyu Lei, Lishi Yang, Liqi Tan, Qibin Yang, Falin Zhou, Shigui Jiang, Jianhua Huang

**Affiliations:** 1Shenzhen Base of South China Sea Fisheries Research Institute, Chinese Academy of Fishery Sciences, Shenzhen 518108, China; 2College of Fisheries and Life Science, Shanghai Ocean University, Shanghai 201306, China; 3Laboratory of South China Sea Fishery Resources Exploitation & Utilization, Ministry of Agriculture and Rural Affairs, South China Sea Fisheries Research Institute, Chinese Academy of Fishery Sciences, Guangzhou 510300, China; 4Shenzhen animal Disease Prevention and Control Center, Shenzhen 518000, China

**Keywords:** *Procambarus clarkii*, air exposure, re-submersion, histological structure, antioxidant activities, gene expression

## Abstract

**Simple Summary:**

Dry transport often leads to air exposure of crayfish, and it is necessary to understand the physiological status of crayfish after air exposure and re-submersion. The aim of this study was to investigate the effect of air exposure on the physiology of crayfish and to evaluate the recovery status of crayfish after re-submersion. Results showed that air exposure had significant effects on the hepatopancreas and gills of crayfish and that re-submersion reduced the harmful response to a certain degree.

**Abstract:**

Air exposure is an important environmental stressor during the transportation and cultivation of *Procambarus clarkii*. We evaluated the effect of re-submersion for 24 h after dry transportation for 24 h on the histological structure, antioxidant activity, and gene expression of crayfish. The antioxidant parameters of catalase (CAT), superoxide dismutase (SOD), malondialdehyde (MDA), and lactate dehydrogenase (LDH), and the relative expression of CAT, SOD, HSP70, and ferritin genes were subsequently measured in the hepatopancreas and gills at both stages. Histopathology found that air exposure led to vacuolation of the hepatopancreas and disorderly arrangement of respiratory epithelial cells (REC) in the gills. The activities of catalase (CAT), superoxide dismutase (SOD), malondialdehyde (MDA), and lactic dehydrogenase (LDH) in the hepatopancreas and gills increased with short-term air exposure. The relative expression of genes (CAT, SOD, HSP70, and Ferritin) were induced after short-term air exposure. During re-submersion, MDA content and CAT and SOD activities in the hepatopancreas and gills were restored after 24 h, however, LDH activity and hepatopancreatic tissue damage were not repaired. Our results indicate that air exposure can cause oxidative damage to *P. clarkii*, and CAT and SOD can be used to determine the response of crayfish exposed to air, in addition to some damage that can be eliminated after re-submersion to a limited degree. This study provides foundational data that re-submersion can improve crayfish performance under hypoxic stress to a certain extent and will lead to the development of more effective transportation strategies and decrease economic losses in the future.

## 1. Introduction

The red swamp crayfish *Procambarus clarkii*, native to north-eastern Mexico and south-central USA, is one of the most valuable freshwater aquaculture species in China [1,2]. *P. clarkii* has many advantages, such as rapid growth, resistance to low oxygen concentrations, and longer survival without water [3,4]. Over the past 10 years, the yield of *P. clarkii* has increased rapidly in China. However, there are many problems with *P. clarkii* culture, such as outdated breeding technology [5], overly clustered culture areas, and degradation of germplasm resources [6] and regular genetic exchanges among the populations are required to address these problems. Most farms use cross-regional introductions to avoid germplasm degradation. Dry transportation is the main method of movement in crustacean culture systems [7,8], including parental and seed-stock transportation, but the approaches often involve air exposure. Most of the transported stock is removed from the water and exposed to air for shipment, which can easily cause crayfish to suffer desiccation and oxidative stress [9,10,11]. The procedure includes transportation, catching, collecting, and delivery. Long-term air exposure may result in irreversible damage to crayfish in subsequent cultures, which is not beneficial for healthy crayfish breeding and increases the cost of cultivation. Previous studies have suggested that desiccation can cause severe metabolic and respiratory disorders [12,13,14], tissue damage [15,16], and apoptosis [17]. Some studies have shown that lowering the temperature can improve the survival rate of crustaceans to a certain extent under desiccation [18]. However, few studies have examined improving transportation quality or explained the mechanism. Understanding the effects of desiccation and re-submersion on the physiological and antioxidant status of red swamp crayfish could improve their transport process and increase the survival rate of subsequent cultures.

Desiccation refers to air exposure that causes respiratory distress or even hypoxic stress in aquatic animals [19]. When crustaceans are subjected to hypoxic stress, they supply energy to the body through anaerobic respiratory metabolism. Crustacean response mechanisms to desiccation are complex, with their regulation involving antioxidant responses [20,21], immune responses [22,23], energy metabolisms [24], and the expression of multiple genes [19,25,26]. In addition to these regulations, crustaceans can also adjust their behavior [10] and decrease their activity [27] to adapt to desiccation conditions. Many studies have shown that antioxidant defense systems in crustaceans, such as superoxide dismutase (SOD) and catalase (CAT) systems, are essential for maintaining the balance of reactive oxygen species (ROS) and preventing damage due to oxidation [28]. Recent studies have shown that the internal microstructure of crustaceans also responds to desiccation stress, such as significant swelling of mitochondria, partial loss of fluid [29], and extensive vacuolization of hepatocytes [15]. Many genes in crustaceans, such as heat shock proteins, also respond to changes in the external environment, including air exposure [19,26] and ammonia exposure [30].

This is based on limited knowledge of the re-submersion phase after transportation. In this study, we simulated short-term dry transport to investigate the changes in antioxidant systems and related tissue structures of *P. clarkii* during air exposure and re-submersion by measuring the antioxidant enzyme activities and gene-related expression levels, as well as observing microstructural changes in the hepatopancreas and gills during the whole process. This study sought to understand the mechanisms of the impact of the transport process on crayfish, and preliminary studies on the recovery state of crayfish after air exposure, in order to provide a reference for improving the mode of transport of crayfish and improving their survival rate.

## 2. Materials and Methods

### 2.1. Animal Materials

Experimental crayfish (body weight: 7 ± 0.5 g) of uniform size and health were quickly selected from a farm in Guangzhou, Guangdong Province, China, in January 2022 on the same day of harvesting, and all crayfish were in the culture pond prior to the experiment. During the culture period, the water pH was 7.6–7.8, and dissolved oxygen levels were 5.0–5.5 mg/L

### 2.2. Experimental Design of Air Exposure and Re-Submersion

The experiments were conducted in a closed, windless laboratory at room temperature (14 ± 2 °C) and relative humidity (65 ± 5%). Considering that with the current limitations in the introduction regions, the time for cross-provincial transportation is generally not more than 24 h, we set the air exposure time to 24 h. To simulate an actual dry transport situation; 180 crayfish were randomly placed in three transport frames with dimensions of 45 × 60 × 30 cm, with 60 crayfish in each group. At 0, 1, 3, 6, 9, 12, and 24 h, four crayfish were randomly selected from each group for analysis. At the end of the 24 h drying period, the remaining crayfish were put back into the water for the next experiment, and four crayfish were randomly selected at 1, 3, 6, 12, and 24 h. In addition, a small amount of hepatopancreas and gill tissue was fixed in a 4% paraformaldehyde solution per group for subsequent histological studies. Crayfish were carefully sampled after hypothermia anesthesia, and the samples were immediately frozen in liquid nitrogen and stored at −80 °C for further enzyme activity and gene expression analysis.

### 2.3. Histopathological Analysis

The hepatopancreas and gills were fixed in 4% paraformaldehyde for more than 24 h, treated with a graded ethanol concentration series for dehydration, and then made transparent with xylene as the transparency medium. This was followed by paraffin immersion and embedding, and after completion of embedding, tissue sections (thickness 5 ± 1 um) were stained with hematoxylin and eosin (H&E. Finally, histopathological assessments were performed using a microscope (Nikon Eclipse E100, Japan).

### 2.4. Determination of Antioxidant Enzyme Parameters

The hepatopancreas and gill samples were weighed and placed in centrifuge tubes, and saline solution (0.85% NaCl) was added at a low temperature (4 °C) at a ratio of 1:9 (w/v). Total protein content was determined, and malondialdehyde (MDA), superoxide dismutase (SOD), catalase (CAT), and lactate dehydrogenase (LDH) activities were measured using the respective kits (MDA assay kit A003-1, SOD assay kit A001-1, CAT assay kit A007-1-1, LDH assay kit A020-1-2, Nanjing Jian Cheng Institute of Biological Engineering, Nanjing, China) according to the manufacturer’s instructions (Nanjing Jian Cheng Institute of Biological Engineering, Nanjing, China).

### 2.5. Gene Expression Analysis

#### 2.5.1. RNA Extraction and cDNA Synthesis

Total RNA was extracted from the hepatopancreas and gills following the instructions of the HiPure Fibrous RNA Plus Kit. RNA purity was tested using a NanoDrop ND-2000, and its integrity was verified using 1% agarose electrophoresis. The extracted total RNA was used as a template for cDNA synthesis according to the instructions of the Prime Script RT Reagent Kit with g DNA Eraser (Takara, Shiga, Japan). Notably, cDNA templates were stored at −80 °C for later analysis.

#### 2.5.2. Real-Time Quantitative PCR

The specific primer pairs of genes (Table 1) were designed using Primer Premier 6.0, based on the references, and sent to the sequencing company for primer synthesis. The 18 s gene was used as an internal control. The cDNA prepared was diluted to (55 ± 2) ng/μL and amplified by RT-PCR using a Roche Light Cycler^®^ 480II (Roche, Germany). Four replicates were set for each sample and an internal reference. The amplification system was 12.5 μL, comprising 1 μL template cDNA, 0.5 μL upstream primer, 0.5 μL downstream primer, 6.25 μL 2X SYBR Green Pro Tap HS Premix, 4.25 μL ddH2O. The reaction conditions were as follows: 95 °C for 30 s; 95 °C for 20 s, 59 °C for 5 s, 45 cycles at 65 °C for 15 s; 55 °C to 97 °C, and 37 °C for 5 min. The 2^−ΔΔCt^ method was used for calculations.

### 2.6. Statistical Analysis

Prior to analysis, all measured variables were checked for normality (Kolmogorov–Smirnov test) and homoscedasticity of variance (Bartlett’s test). If these conditions were satisfied, ANOVE and Tukey tests were used to determine the differences between the groups. In other situations, non-parametric Kruskal-Wallis tests were performed. The null hypothesis was rejected at α < 0.05 in all tests. All data are presented as the mean ± SD.

## 3. Results

### 3.1. Histological Changes of Hepatopancreas and Gills after Air Exposure and Re-Submersion

The hepatopancreas and gills of crayfish were stained with HE eosin. The histopathological characteristics of the hepatopancreas and gills of air-exposed and re-submerged crayfish are shown in Figure 1 and Figure 2**,** respectively. Histomorphological characteristics of the hepatopancreas and gills of air-exposed and re-submerged crayfish are shown in Table 2. The hepatic ducts of the hepatopancreas were tightly arranged and had a normal cell structure (-) at 0 h. The hepatopancreas of air-exposed crayfish showed vacuolization (+) from 1 h to 24 h and gradually increased (+++), and the lumen gradually became larger (++) with increasing air exposure time. The number of B cell transit vesicles (++) increased after exposure to air for 6 h. The number of hepatopancreatic R-cells decreased (+) during air exposure. No significant improvement in vacuolization (++) was observed during the re-submersion phase, the lumen (+) did not improve with longer re-submersion time, and the same was not true for R-cells (+) and B cells (+).

At 0 h, the gill membrane was intact (-), the structure of the respiratory epithelial cells (REC) was clear (-), and there were more hemocytes (-). At 3 h, gill filaments were loosely connected (+). With the extension of air exposure time, after air exposure for 3 h, most of the REC (+) were loosely arranged and detached. After 9 h of air exposure, the number of hemocytes decreased (++), the gill membrane was severely damaged (++), and the normal gill cell structure was lost. After re-submersion, REC remained loosely arranged (+++) and did not increase significantly. The number of hemocytes showed signs of increase (++) with prolonged submersion time from R-3 h, but generally, the gills did not return to the normal state.

### 3.2. Antioxidant and Metabolic Enzyme Changes after Air Exposure and Re-Submersion

Changes in antioxidant and metabolic enzymes after air exposure and re-submersion were observed (Figure 3 and Figure 4).

In the hepatopancreas, CAT activity increased until 9 h, reached a maximum at 9 h (*p* < 0.05), increased again after R-1 h, and then recovered to a level close to that of the initial time (Figure 3A). SOD activity showed an overall increase and decreased at 9 h (Figure 3B). The MDA content in the hepatopancreas increased with the extension of air exposure time and began to decrease after R-6 h in re-submersion (Figure 3C). LDH enzyme activity also increased after air exposure and fluctuated during re-submersion (Figure 3D).

In the gills, CAT activity increased and reached its highest at air exposure 1 h (*p* < 0.05) (Figure 4E) and increased at re-submersion as well. SOD activity significantly increased at 6 h (*p* < 0.05) (Figure 4F) and then recovered to a level close to that of the initial time after re-submersion. MDA content increased, reaching a maximum at 6 h and R-3 h (*p* < 0.05) (Figure 4G). LDH activity increased at 6 h and then decreased to lower levels at re-submersion consistently. (Figure 4H).

### 3.3. Gene Expression in the Hepatopancreas and Gills

Changes in the relative expression of genes during air exposure and re-submersion are shown in Figure 5 and Figure 6.

In the hepatopancreas, the relative expression of the CAT gene reached the highest level at 3 h (*p* < 0.05) and then decreased to the lowest level at 12 h (*p* < 0.05) (Figure 5A); similar changes were observed at re-submersion. The relative expression of the SOD gene increased, reaching the highest level at 3 h (*p* < 0.05), then decreased (Figure 5B), and recovered to a level close to the initial time at re-submersion. Relative expression of the HSP70 gene generally increased, reaching its highest level at 12 h (*p* < 0.05), and increasing to higher levels in R-6 h at re-submersion (*p* < 0.05) (Figure 5C). Relative expression of the ferritin gene in the hepatopancreas reached its highest level at 3 h (*p* < 0.05), and similar changes were observed at re-submersion, reaching its highest level at R-12 h (Figure 5D).

In the gills, the relative expression of the CAT gene showed similar changes in the air exposure and re-submersion phase, each reaching higher at 1 h and highest R-6 h (*p* < 0.05) (Figure 6E). The relative expression of the SOD gene decreased to below the initial level before 12 h of air exposure and then increased after 12 h. At re-submersion, the relative expression of the SOD gene increased and then decreased, eventually below the level of the initial time (Figure 6F). Relative expression of the HSP70 gene generally increased with the extension of air exposure time, reaching its highest at 24 h (*p* < 0.05) (Figure 6G)), and similar changes were observed at re-submersion and at R-24 h decreased below the initial time. The relative expression of the ferritin gene increased to a maximum after 24 h (*p* < 0.05) (Figure 6H), gradually returning to the initial level after rising at re-submersion R-12 h.

## 4. Discussion

Although crayfish have a certain ability to tolerate air exposure, the organism needs to use the water preserved in the gills to maintain its normal respiratory metabolism. Once the water in the gills is depleted, its respiratory metabolism is disrupted, affecting the antioxidant and immune status and causing tissue damage [12,28,32]. The effects of air exposure on the antioxidant capacity of aquatic animals have been extensively studied. The SOD activity and T-AOC levels of Chinese mitten crabs significantly increased at 6 h under air exposure [33]. Reducing the negative impacts of transportation and the expansion of crayfish aquaculture areas has become the leading problem to be solved in the crayfish breeding industry. In this study, we attempted to address the damage caused by the dry transportation process through re-submersion of the crayfish and produced evidence of the histological and physiological changes during the entire process.

During dry transportation, crustaceans are exposed to hypoxia, high temperatures, irradiation, and sudden changes in pH that can lead to histopathological and ultrastructural alterations. Histological and ultrastructural observations are standard methods for pathological examination and are useful tools for assessing the health status of crustaceans [34]. Previous studies have shown that the lumen of hepatopancreatic tubules of *Litopenaeus Vannamei* under cyclic serious/medium hypoxic conditions appear expanded on day 1, vacuolated on day 3, and gradually the dispersive vacuoles gathered into large vacuoles [35]. The hepatopancreatic tubules of the hepatopancreas also showed serious inflation and lumen enlargement under air exposure in *Penaeus monodon* [16]. The present study also found that vacuolization occurred after 1 h of air exposure and that the lumen of hepatopancreatic cells gradually became larger and more vacuolated as the air exposure time increased. After 6 h of air exposure, the number of B cell transit vesicles increased and their volume increased, which may contribute to the metabolic absorption of nutrients in the hepatic tubules and provide more energy [36]. The main function of R-cells is to store nutrients [37], and the number of R-cells decreases with longer air exposure times, indicating that crayfish physiologically regulate by consuming their own stored nutrients. The absence of lumen alleviation and vacuolation after re-submersion suggests that damage to the hepatopancreas from air exposure is difficult to restore within a short period. 

In addition, the gills of crayfish showed damage with RECs loosely arranged at 3 h, and loosely attached to the filaments until 24 h with loss of their normal structure, most of which were detached and lost their normal structure. Damage to the gills may have led to a decrease in the oxygen acquisition capacity of crayfish, weakening aerobic metabolism and enhancing anaerobic metabolism due to gill hypoxia [37]. In studies of *Macrobrachium nipponense* [38] and *Procambarus clarkii* [37], the disorganization, separation, and detachment of the REC arrangement were also found. After re-submersion, the lumen and vacuolation of the hepatopancreas showed little recovery, and the gill structure did not improve significantly, indicating that tissue damage is difficult to reverse in a short period and that the hepatopancreas has a better self-repair ability than the gills. After a short period of dry transportation, the tissue structure of crayfish will inevitably suffer some damage. Although the effects due to air exposure after re-submersion may not be serious, it will undoubtedly still affect its normal physiological function.

When crustaceans are exposed to extreme environments, drastic changes in dissolved oxygen, temperature, and osmotic pressure may cause a combined effect of oxidative stress, leading to ROS production. ROS are involved in oxygen-sensing mechanisms and antioxidant systems and are generally considered harmful to living organisms [39,40]. Antioxidant enzymes such as superoxide dismutase (SOD) and catalase (CAT) are essential for eliminating excess ROS and preventing oxidative damage [41]. In the present study, CAT and SOD activity increased quickly at 3 h and finally decreased at the end of stress. Similar findings were found in studies of the effects of air exposure on *Procambarus clarkii* [29,32] and *Exopalaemon carinicauda* [42]. Air exposure triggers the antioxidant system in the hepatopancreas and gills, increasing CAT and SOD activity to eliminate excess ROS. Subsequent decreases in both CAT and SOD activities imply a reduced ability to scavenge ROS and may increase the risk of oxidative damage to cellular components. Notably, the relative expression of CAT and SOD genes in the gills was not synchronized with the changes in CAT and SOD activity during the air exposure stage, which may be explained by the fact that HSP70 can release and directly increase endogenous superoxide oxidase activity to mitigate oxidative damage [18]. During the re-submersion stage, CAT activity in the hepatopancreas increased significantly and then decreased rapidly to that of the initial time of re-submersion, whereas CAT activity in the gills increased with extended re-submersion time. In addition, SOD activity in the hepatopancreas was always lower than the initial time during the re-submersion stage, while SOD activity in the gills increased with extended re-submersion time. This indicates that the antioxidant enzyme SOD was consumed in large amounts during the re-submersion stage, and when the organism could not provide sufficient antioxidant enzymes, more antioxidant substances were needed to resist oxidative stress and maintain the balance of ROS. Compared to the hepatopancreas, the gill, as an exposed respiratory organ, is in direct contact with the outside world and responds relatively quickly to environmental changes [43], as evidenced by the changes in CAT and SOD activity in the gills. This also implies that CAT and SOD activities may be indicators of response to air exposure stress. Taken together, these results suggest that re-submersion significantly relieves antioxidant stress, but the crayfish do not completely recover in the short term.

MDA is the end-product of lipid peroxidation, which causes toxic stress in cells and is used as a biomarker to measure the level of oxidative stress in an organism [44]. In the present study, MDA content in the hepatopancreas and gills increased with extended air exposure time, and the peak value appeared later than that of CAT and SOD, confirming that it is a byproduct of ROS. Previous studies have also reported that air exposure increases MDA levels in the hepatopancreas of *Litopenaeus vannamei* [20]. At the beginning of air exposure, the excessive ROS exceeded the antioxidant defense capacity, and with the accumulation of reactive oxygen species, the activation of the antioxidant system enhanced the ability to scavenge ROS and decrease MDA content, indicating that air exposure interferes with the oxidative balance in the hepatopancreas and gills of crayfish. In addition, MDA content increased significantly (*p* < 0.05) in the hepatopancreas and gills during the re-submersion phase, reaching the highest values at 12 h and 3 h, respectively (*p* < 0.05), and then decreased to the initial level, indicating that crayfish have a strong resistance to oxidative stress. The results of Wang et al. [32] also proved this [32]. MDA is positively correlated with SOD level, and a recovery state of MDA revealed alleviation of the oxidation reaction and repair of cell damage by re-submersion in a water environment. This indicates that MDA may be a valid indicator for assessing the degree of health recovery of crayfish during re-submersion.

LDH is a marker enzyme of anaerobic respiratory metabolism, which can catalyze the conversion of pyruvate to lactate, and the decomposition of lactate provides ATP for the body. Its activity can reflect the capacity of anaerobic metabolism to a certain extent, which is a useful indicator to evaluate the capacity of the anaerobic metabolism [12,45,46]. It was previously reported that the LDH activity of *Exopalaemon carinicauda* muscle increases during air exposure to eliminate the accumulation of lactic acid caused by the increased metabolic capacity of anaerobic respiration and subsequently decreases due to the conversion of lactic acid to maintain energy supply [47]. In the present study, LDH activity in the hepatopancreas and gills of crayfish increased significantly (*p* < 0.05) after 1 h of air exposure, indicating that the intensity of anaerobic metabolism rapidly increased to satisfy the energy requirements under air exposure conditions, followed by a similar change of the decrease in the LDH activity. The reason for this, besides the maintenance of the energy supply, is presumably that the metabolic system of the organism is affected by the excessive duration of stress. LDH activity in the hepatopancreas decreased and then increased, indicating that the organism was still undergoing anaerobic metabolism. The hepatopancreas is the digestive and metabolic center of shrimp and has high levels of metabolic enzyme activity, a relatively high metabolic rate, and is more susceptible to oxidative stress during air exposure [48]. When re-submerged, the LDH activity in the hepatopancreas and gills showed slight fluctuations, returning to a level similar to the initial time, which could indicate that aerobic respiration had recovered at submersion and that the gills responded more rapidly than the hepatopancreas. The results also demonstrated signs of recovery in crayfish physiological indicators after re-submersion.

Heat shock proteins are involved in innate and adaptive immunity, as well as in antioxidant processes, and HSP70 is a member of the heat shock protein family that plays an important role in protecting organisms from the harmful effects of stressors such as heat and cold shock and low oxygen levels [49,50]. The HSP70 gene has been studied extensively. HSP70 plays an important role in the protection of fish, shrimp, and mollusks from biotic and abiotic stresses that may be related to cellular homeostasis, including the proper folding, localization, and degradation of proteins [51]. The expression of HSP70 was significantly altered in both the hepatopancreas and gills of *Rachycentron canadum* under hypoxic stress [52]. The relative expression of Hsp70 in the hemolymph of *Eriocheir sinensis* decreased significantly under acute hypoxic conditions [53]. In the present study, the relative expression of the HSP70 gene in the hepatopancreas and gills of crayfish also showed a significant increase (*p* < 0.05) under air exposure, reaching the highest levels at 12 h and 24 h, respectively, and a more stable increase in the gills. The expression of the HSP70 gene showed a fluctuation during the early stages of the re-submersion phase and finally reached normal levels at 24 h, but recovery in the gills seemed to be slow. HSP70, as a protective protein in the hepatopancreas, is the first to be upregulated compared to other tissues to eliminate damage to the hepatopancreas from environmental changes, which is similar to the findings reported in other studies [54].

Ferritin is a protein that stores iron and is widely found in various organisms. It is able to resist oxidative damage from environmental stress, plays an important role in regulating the balance of iron metabolism, detoxification, and antioxidation [55,56], and can reduce the accumulation of ROS in response to oxidative stress [57]. In previous studies, HSP70 and ferritin were found to play a synergistic role as antioxidants, against the damage caused by desiccation stress in *Penaeus monodon* [16]. In the present study, the relative expression of the ferritin gene showed a similar fluctuation trend to HSP70, confirming a potential synergistic function between the two genes. In detail, the relative expression of the ferritin gene showed a very high level at the end of stress in gills and recovered to a common state after 24 h of re-submersion, and it was speculated that the long-term loss of water in gills meant the ferritin gene was rapidly upregulated to eliminate oxidative damage. When crayfish leave the water, the environment changes, but some water remains in the gills, temporarily maintaining the physiology of the body in dynamic equilibrium. When the water is completely lost, the exposure of crayfish to the air suddenly reduces the oxygen supply, resulting in the production of oxygen radicals in the crayfish, inducing the rapid upregulation of the ferritin gene to eliminate oxidative damage caused by oxygen radicals. In addition, the decrease in the relative expression of ferritin genes may be the result of the decreased appearance of oxygen radicals. Moreover, compared with HSP70, it can be seen that ferritin genes play a secondary role in antioxidant response. The relative expression of ferritin genes in the hepatopancreas fluctuated greatly after re-submersion, decreased at 24 h, and was significantly lower than that at the initial time. Combined with structural changes in the hepatopancreas, this may be because at the beginning of the submersion stage, the metabolism of the crayfish organism was still disturbed, causing stress damage [58], and the expression of the ferritin gene was inhibited. The relative expression of ferritin genes in gills could be restored to the initial level after 24 h of submersion, unlike the hepatopancreas, implying that gills are more resilient. This may be due to the fact that the hepatopancreas plays a key role in iron metabolism, storing excess iron to inhibit Fenton’s reaction and achieve an antioxidant effect [59] rather than being more sensitive to oxidative stress.

## 5. Conclusions

In conclusion, air exposure altered the histological structure, antioxidant status, and expression of related genes and triggered the antioxidant responses of several enzymes in *P. clarkii*. This suggests that the antioxidant system is activated to defend against air exposure, and CAT and SOD might be effective indicators for monitoring air exposure in crayfish. When crayfish return to the water environment, the antioxidant status of the organism shows signs of recovery, during which MDA may be used as an indicator to assess the degree of crayfish recovery; however, tissue damage to the organism caused by air exposure is difficult to reverse in a short period. Therefore, dry transport is not suitable as the main mode of transportation for crayfish aquaculture, especially when cross-regional introduction inevitably affects the physiological state of crayfish, which in turn affects the subsequent parental breeding, resulting in significant losses. The results of this study provide valuable data on the dry transport and regulatory mechanisms of resistance to air exposure and will benefit the development of more appropriate transport strategies in breeding practice.

## Figures and Tables

**Figure 1 animals-13-00462-f001:**
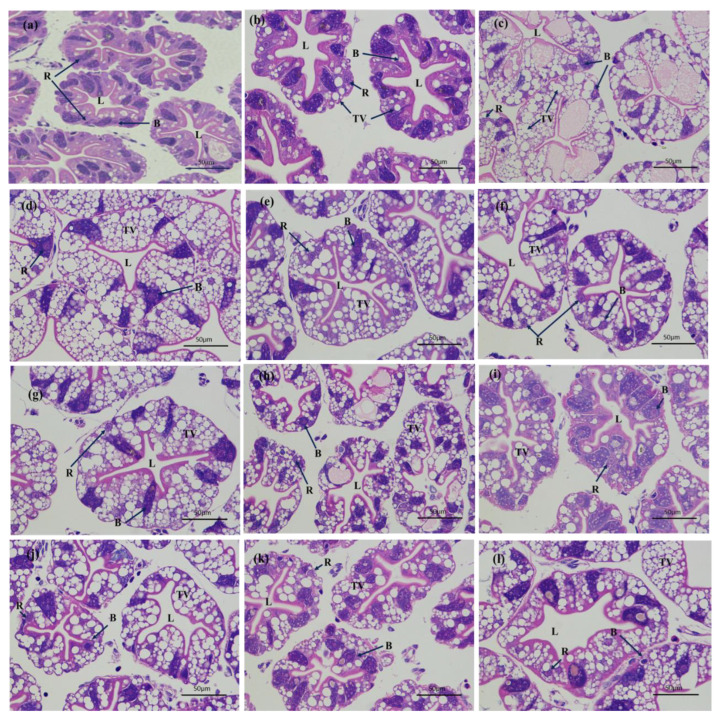
Effect of air exposure on hepatopancreas microstructure of *P. clarkia.* Notably, (**a**–**g**) show air exposure of 0 h, 1 h, 3 h, 6 h, 9 h, 12 h, and 24 h, respectively. Moreover, (**h**–**l**) shows re-submersion of 1 h, 3 h, 6 h, 12 h, and 24 h, respectively. B. B cell; R. R cell; L. Lumen; TV. Transferred vacuoles.

**Figure 2 animals-13-00462-f002:**
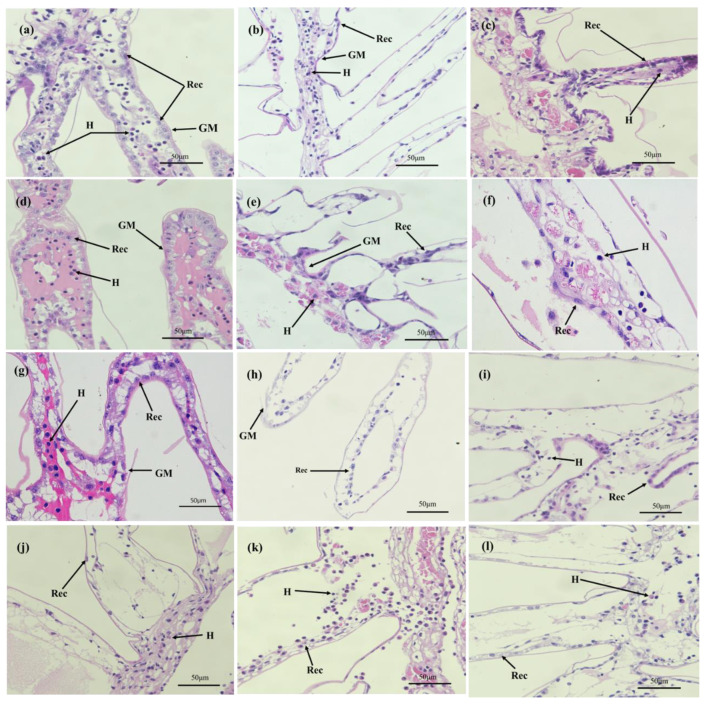
Effect of air exposure on gills microstructure of *P. clarkii.* (**a**–**g**) show air exposure of 0 h, 1 h, 3 h, 6 h, 9 h, 12 h, and 24 h, respectively. (**h**–**l**) show re-submersion of 1 h, 3 h, 6 h, 12 h, and 24 h, respectively. Rec. Respiratory epithelium cells; H. Haemocytes; GM. Gill membranes.

**Figure 3 animals-13-00462-f003:**
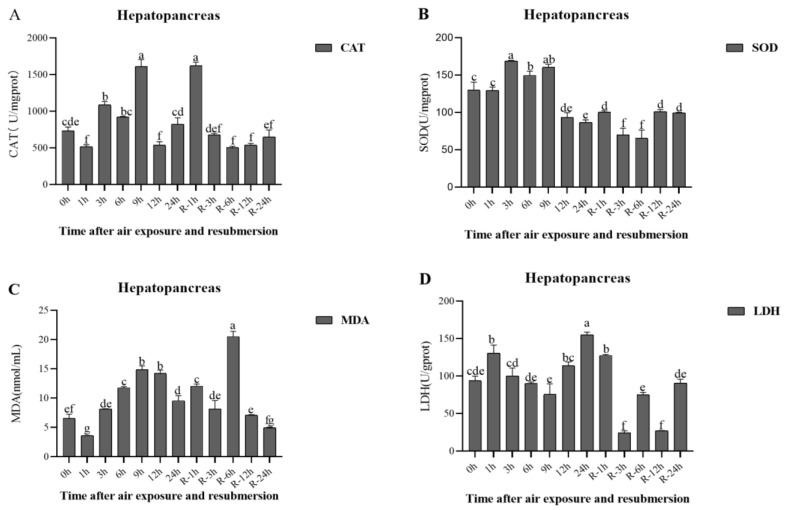
CAT activity (**A**), SOD activity (**B**), MDA content (**C**), and LDH activity (**D**) in the hepatopancreas of *P. clarkii* after air exposure for 0 h, 1 h, 3 h, 6 h, 9 h, 12 h and 24 h, including re-submersion for R-1 h, R-3 h, R-6 h, R-12 h, R-24 h. Different letters show significant differences between different exposure times in the same tissue (*p* < 0.05).

**Figure 4 animals-13-00462-f004:**
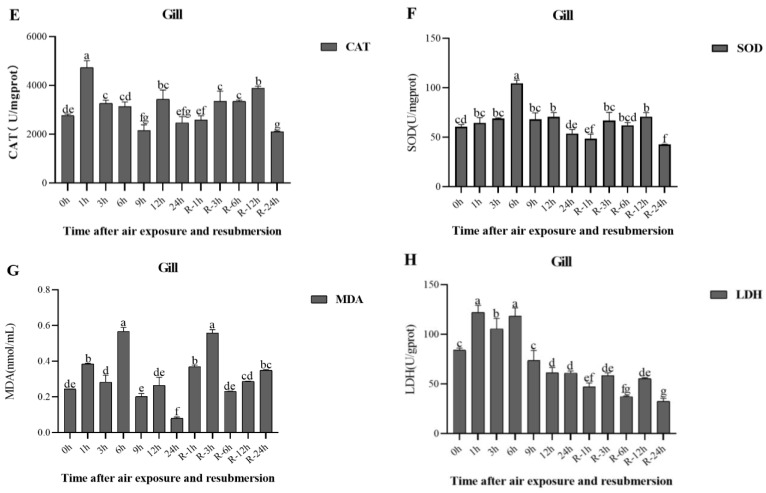
CAT activity (**E**), SOD activity (**F**), MDA content (**G**), and LDH activity (**H**) in the gill of *P. clarkii* after air exposure for 0 h, 1 h, 3 h, 6 h, 9 h, 12 h and 24 h, including re-submersion for R-1 h, R-3 h, R-6 h, R-12 h, and R-24 h. Different letters show significant differences between different exposure times in the same tissue (*p* < 0.05).

**Figure 5 animals-13-00462-f005:**
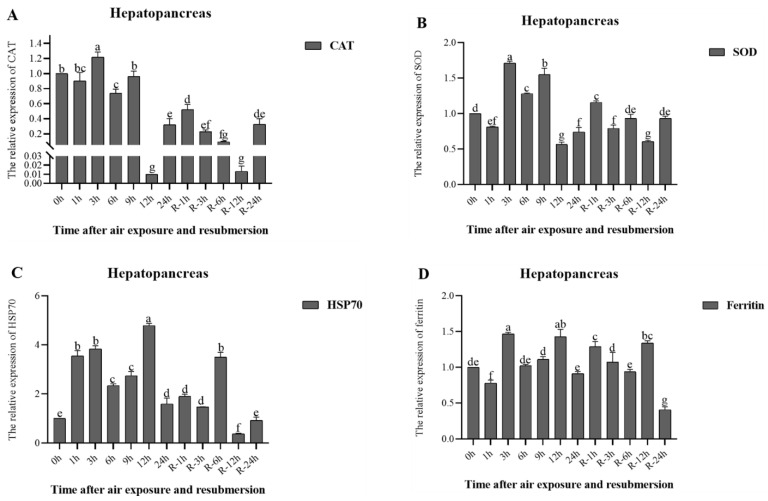
Relative expression levels of CAT (**A**), SOD (**B**), HSP70 (**C**), and Ferritin (**D**) in the hepatopancreas of *P. clarkii* after air exposure for 0 h, 1 h, 3 h, 6 h, 9 h, 12 h, and 24 h, including re-submersion for R-1 h, R-3 h, R-6 h, R-12 h, and R-24 h. Different letters represent significant differences between various exposure times (*p* < 0.05).

**Figure 6 animals-13-00462-f006:**
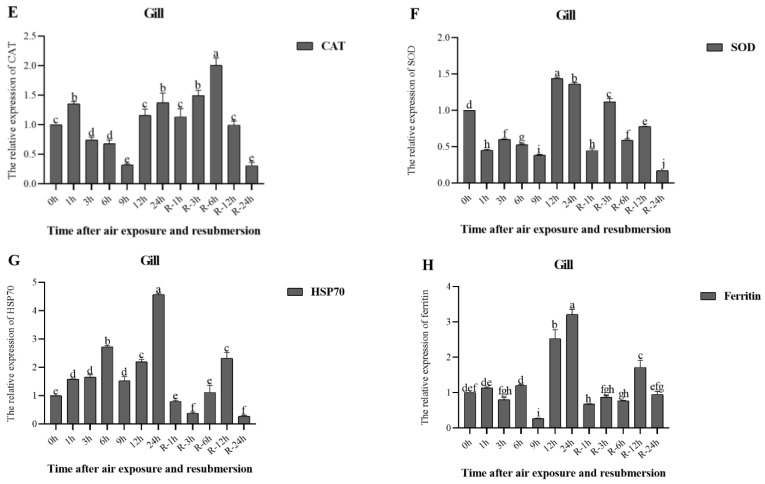
Relative expression levels of CAT (**E**), SOD (**F**), HSP70 (**G**), and Ferritin (**H**) in the gills of *P. clarkii* after air exposure for 0 h, 1 h, 3 h, 6 h, 9 h, 12 h, and 24 h, including re-submersion for R-1 h, R-3 h, R-6 h, R-12 h, and R-24 h. Different letters represent significant differences between various exposure times (*p* < 0.05).

**Table 1 animals-13-00462-t001:** Primer pairs used in qPCR.

Primer Name	Nucleotide Sequence (5′-3′)
18S-F	TCTTCTTAGAGGGATTAGCGG
18S-R	AAGGGGATTGAACGGGTTA
CAT-F	GCTGAGGTGGAACAGATGGCAAT
CAT-R	CGATGAGTGTCATTGTAGGCGAAGA
SOD-F	GAGGCAGACTACCAAGGA
SOD-R	ATGGACAACGATGGCTAG
HSP70-F	TCAGCATCAAGTCGGCAGTCTCT
HSP70-R	TCCTTCATCTGGTGCTCGTATTCCT
Ferritin-F	ATCCGCCAGAACTACCAT
Ferritin-R	TTCACGCTCTTCATCACTT

**Table 2 animals-13-00462-t002:** Histomorphology characters in hepatopancreas and gills of air-exposed and re-submersion crayfish.

		0 h	1 h	3 h	6 h	9 h	12 h	24 h	R-1 h	R-3 h	R-6 h	R-12 h	R-24 h
Hepatopancreas	L	(-)	(+)	(+)	(++)	(++)	(++)	(++)	(++)	(+)	(+)	(+)	(+)
	B	(-)	(+)	(+)	(++)	(++)	(++)	(++)	(++)	(+)	(+)	(+)	(+)
	R	(-)	(-)	(+)	(+)	(+)	(+)	(++)	(++)	(+)	(+)	(+)	(+)
	TV	(-)	(+)	(++)	(+++)	(+++)	(+++)	(+++)	(+++)	(++)	(++)	(++)	(++)
Gill	Rec	(-)	(-)	(+)	(+)	(++)	(++)	(+++)	(+++)	(+++)	(++)	(++)	(++)
	H	(-)	(-)	(-)	(-)	(++)	(++)	(++)	(+++)	(+++)	(++)	(++)	(++)
	GM	(-)	(-)	(+)	(+)	(++)	(++)	(++)	(++)	(++)	(++)	(++)	(++)

L. Lumen; B. B cell; R. R cell; TV. Transferred vacuoles. Rec. Respiratory epithelium cells; H. Hemocytes; GM. Gill membranes. Note: Pathology was scored as (-) no pathology; (+) pathology in < 30% of fields; (++) pathology in 30–70% of fields, and (+++) pathology in > 70% of fields [31].

## Data Availability

Data are contained within the article.

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
