# Peer review of "Effect of Air Exposure and Re-Submersion on the Histological Structure, Antioxidant Response, and Gene Expression of Procambarus Clarkii"

_animals, 2023, doi:10.3390/ani13030462_

Round 1
Reviewer 1 Report
General comments
This study deals with the effect of air exposure and re-submersion on the histological structure, antioxidant responses of several enzymes and the expression of related genes of the crayfish, Procambarus clarkii for improving the method of dry transport.
Time design of experiment is appropriate. I cannot judge a validity about temperature design of experiment. More explanation on the transport temperature in introduction section.
Some descriptive are not agreement with the result and discussion sections.
I cannot understand sufficiently about histological structure, because of letters in photographs is indistinct.
More explanation on value at 0h in Figs. 4 and 6. These values are same at 0h or 24 h in Figs. 3 and 5.
In conclusions section, authors described ‘the tissue damage caused by air exposure is difficult to reverse in a short period of time, may eventually lead to death.’
If crayfish dead, some parameter (CAT and SOD activity, and MDA) will not be helpful as valid indicator.
I'm doubtful whether these results are practical useful for improving the method of dry transport.
Does authors propose for improving methods the survival rate of crayfish in the transport?
I think that the more appropriate aim of this study is to elucidate the cause of death of crayfish in dry transport.
This text is unclear on the whole. For example, L267. The manuscript would be improved by either a native English speaker or a professional editor.
Therefore, I cannot recommend publication of this manuscript.
Specific comments
Introduction
Authors should introduce the reference no. 30 such as related investigation deal with Procambarus clarkii.
Line number 72 The reactive oxygen species (ROS) should be shown here.
Materials and Methods
Line number 102 I don’t know whether this temperature is a proper.
More explanation on the transport temperature (normal temperature or cool) or transport season (summer, winter or all the year round) in introduction section.
Results
The survival rate of a crayfish should be shown. At least, end of air exposure and re-submersion tests.
More explanation on microstructure in Figs. 1 and 2.
Although the authors describe B-cell in discussion section, not write in results section (see Line number 258-259).
A descriptive of R-cell is not in agreement by the result (Line number 155-156) and discussion (Line number 261).
Line number 182-183 More explanation on LDH enzyme activity (see Line number 338-339).
Line number 207 Is it correct to describe as ’reaching the highest level at 3h and 12h, respectively.’ ?
Line number 209-210 More explanation on the relative expression of ferritin gene (see Line number 381-383).
Discussion
Line number 266 Authors described the gills of crayfish showed slight damage at 1 h of air exposure. However authors has not describe it in results section.
Line number 267 The words (at 3h) overlaps.
Line number 325 Is it correct to describe as ’ In addition, the MDA content increased significantly (P < 0.05) in the hepatopancreas and gills during the re- submersion phase, reaching the highest values at 6 h and 3 h, respectively (P < 0.05)’ ? Because of re- submersion phase and not air exposure phase.
Line number 365-366 Is it correct to describe as ’ Relative expression of HSP70 is known as a potential stress marker in crustaceans [52].’ ?
Line number 368 How did you observe the protect cell?
Line number 385-387 I cannot understand this text to an exact.
Figs. 1 and 2 Authors should make letters in photographs clearer.
Figs. 4 and 6 More explanation on 0h on x-axis label.
Author Response
To REVIEWER General comments 1. about temperature design of experiment. Response: Thank you for your suggestion. Temperature is an important environmental factor for crustaceans, and considering that low temperatures are beneficial for shrimp and crab transport, this experiment was designed to be conducted in January with an actual temperature of 14±2°C. Related studies are added in the introduction. 2. Some descriptive are not agreement with the result and discussion sections Response: We are very sorry for this error. The errors have been corrected in the article, e.g., R-cell in the histology section. 3. about histological structure, because of letters in photographs is indistinct. Response: Sorry about this, we have zoomed in on the picture letters. 4. More explanation on value at 0h in Figs. 4 and 6. These values are same at 0h or 24 h in Figs. 3 and 5. Response: Fig3, Fig4, Fig5, Fig6 have been remade in the article, and will be put together with the same tissue, more convenient for you to check. 5. In conclusions section, authors described ‘the tissue damage caused by air exposure is difficult to reverse in a short period of time, may eventually lead to death.’ If crayfish dead, some parameter (CAT and SOD activity, and MDA) will not be helpful as valid indicator. Response: Thank you for the reminder. This study found that crayfish have difficulty recovering from tissue damage within a short period of time due to air exposure. The " may eventually lead to death " is based on the damage to crayfish caused by air exposure, which is presumed to occur with prolonged air exposure, and some studies have shown crayfish death after 24H of air exposure (DOI: 10.12131/20190059). To avoid misunderstanding, we will remove the conclusion that it can lead to death. 6. I'm doubtful whether these results are practical useful for improving the method of dry transport. Response: Thank you for your doubt. Through the 24H air exposure experiment, we can observe that around 3H, air exposure has little effect on the physiological function of crayfish, and beyond 3H, transportation methods should be considered in other ways. 7. Do authors propose for improving methods the survival rate of crayfish in the transport? Response: Thank you for your question. How to improve the survival rate of crayfish in transport is a good research direction for us. At present, we mainly focus on the recovery effect of re-submersion on crayfish transport. As mentioned in the introduction, low temperature to improve the survival rate of crayfish transport may be a way. 8. I think that the more appropriate aim of this study is to elucidate the cause of death of crayfish in dry transport. Response: Thank you for your suggestion. Due to air exposure for 24 h, the crayfish were weak in activity but did not appear to die. We will consider extended air exposure time to discover the exact cause of crayfish death due to dry transport. 9. This text is unclear on the whole. For example, L267. The manuscript would be improved by either a native English speaker or a professional editor. Response: We are very sorry for this. We have corrected the inappropriate sentence. Specific comments Introduction 1. Point 1: Authors should introduce the reference no. 30 such as related investigation deal with Procambarus clarkii. Response 1: Thank you for your suggestions. The no. 30 reference provides us with important reference value, and the findings of its antioxidant section are important for us. However, the experimental design and the focus of the study were different, so we did not present them as detailed in the article. 2. Point 2: Line number 72 The reactive oxygen species (ROS) should be shown here. Response 2:Thank you for the reminder. It has been changed in the article. Materials and Methods 1. Point 3: Line number 102 I don’t know whether this temperature is a proper. Response 3: Sorry for the disturbance. At this temperature, shrimp and crabs are less likely to fight and avoid damage. And low temperatures are beneficial for shrimp and crab transport. 2. Point 4: More explanation on the transport temperature (normal temperature or cool) or transport season (summer, winter or all the year round) in introduction section. Response 4: Thank you for your advice. In southern China, the actual temperature in January is around 15°C, and some studies have shown that low temperature air exposure has a higher survival rate than room temperature(DOI:10.16378/j.cnki.1003-1111.19228). In addition, in the southern aquaculture region of China, farmers often transport in the spring when the temperature is low, for one thing, the temperature is low and the aquatic products are not as vigorous to avoid damage, and for another, they can extend the growth time (DOI: 10.3969/j.issn.1002-6681.2006.03.025). Results 1. Point 5: The survival rate of a crayfish should be shown. At least, end of air exposure and re-submersion tests. Response 5: Thank you for your reminder. The survival rate during air exposure and re-submersion in this experiment was 100%, and mortality could occur with extended air exposure. And some studies have shown that under room temperature conditions, crayfish die from air exposure for 24 h, and the mortality rate can exceed 15%https://doi.org/10.1016/j.aqrep.2021.100898. 2. Point 6: More explanation on microstructure in Figs. 1 and 2. Response 6: Thank you for your advice. The microstructural changes in hepatopancreas and gill have been further described in the article. Supplemented with changes during re-submersion. 3. Point 7: Although the authors describe B-cell in discussion section, not write in results section (see Line number 258-259). Response 7: We are sorry for this problem. We have made a change in the article and added the description of the B-cell in the results section of the article. 4. Point 8: A descriptive of R-cell is not in agreement by the result (Line number 155-156) and discussion (Line number 261). Response 8: We are sorry for this problem. In the results, the number of R-cells decreased with increasing air exposure time, which has been changed in the article. 5. Point 9: Line number 182-183 More explanation on LDH enzyme activity (see Line number 338-339). Response 9: We are sorry to have this problem. LDH enzyme activity in the hepatopancreas and gills also rises during the air exposure reaches a maximum at 24h and 3h, respectively. And both begin to decline during the re-submersion phase. 6. Point 10: Line number 207 Is it correct to describe as ’reaching the highest level at 3h and 12h, respectively.’ ? Response 10: Relative expression of SOD gene in the hepatopancreas and gills increased, reaching the highest level at 3 H and 12 H, respectively. 7. Point 11: Line number 209-210 More explanation on the relative expression of ferritin gene (see Line number 381-383). Response 11: In detail, ferritin mRNA showed a very high level at the end of stress in gill and recover to a common state after 24 h re-submersion, it speculated that along time loss of water in gills inducing ferritin gene rapidly upregulated to eliminate oxidative damage. Discussion 1. Point 12: Line number 266 Authors described the gills of crayfish showed slight damage at 1 h of air exposure. However, authors have not described it in results section. Response 12: Sorry to have such problems. The changes in the crayfish gills were more obvious in 3h, showed damage with the Rec were loosely arranged, and loosely attachment of the filaments. Therefore, we removed the state of 1h from the article. 2. Point 13: Line number 267 The words (at 3h) overlaps. Response 13: Excess words have been deleted. 3. Point 14: Line number 325 Is it correct to describe as’ In addition, the MDA content increased significantly (P < 0.05) in the hepatopancreas and gills during the re- submersion phase, reaching the highest values at 6 h and 3 h, respectively (P < 0.05)’ ? Because of re- submersion phase and not air exposure phase. Response 14: I regret that the results may have been misinterpreted. The temporary increase in MDA content during re-submersion may be caused by the increase in oxidative substances accumulated in the organism, when CAT and SOD activities were low, followed by a decrease in MDA content in liver and gills, indicating that the antioxidant function of crayfish recovered after re-submersion and was able to reach its balance. 4. Point 15: Line number 365-366 Is it correct to describe as’ Relative expression of HSP70 is known as a potential stress marker in crustaceans [52].’ ? Response 15: Thank you for the reminder. There is indeed a problem with that description. We will highlight the role of HSP70. HSP70 plays an important role in the protection of fish, shrimp and mollusks from biotic and abiotic stresses that may be related to cellular homeostasis, including proper folding, localization and degradation of proteins. 5. Point 16: Line number 368 How did you observe the protect cell? Response 16: We are sorry to have disturbed you. It is perhaps inappropriate to describe it that way; the present experiment really did not explore it at the cellular level. To avoid misunderstanding, we will delete this sentence. 6. Point 17: Line number 385-387 I cannot understand this text to an exact. Response 17: Sorry for the confusion. In short, what we want to express is the relative expression of ferritin gene showed a very high level at the end of stress in gill. 7. Point 18: Figs. 1 and 2 Authors should make letters in photographs clearer. Response 18: Thank you for the reminder. We have zoomed in on the pictures. 8. Point 19: Figs. 4 and 6 More explanation on 0h on x-axis label Response 19: Thank you for the reminder. We have Fig4, Fig6 of the hepatopancreas and gills indicated separately and 0 h indicated the initial time.
Reviewer 2 Report
The manuscript #animals-2067122 investigates an important and interesting topic of metabolic disturbances in a cultured crayfish during relatively short transportation without water. The chosen methodological approach is relevant, and the presented experiment seems to be well organized. I must mention that I’m not a specialist in histology and my review of the respective section is modest. To my opinion, the study deserves publication in Animals, but the following issues should be addressed before that:
Major concerns:
1) My main concern about the current version of manuscript is statistical analysis for the biochemical and expression measurements. I appreciate that the authors used 12 biological replicates in each timepoint of the experiment (4 animals from each of 3 tanks; seems to be quite fine) and the Tukey’s test, which intrinsically takes into account the correction for multiple comparisons. However, 12 measurements in each group is not really enough for robust checking of the data normality (as well as homoscedasticity), which is a very basic requirement for performing ANOVA and Tukey’s test. I understand that SPSS certainly performs the relevant tests (and may even say that normality assumption is not violated) before making ANOVA and further post-hoc tests, but 12 samples is just not enough to make the tests robust enough. Usually, 20 measurements are suggested for such normality tests at the very least. Furthermore, the differences in standard deviations between separate timepoints on Figs 3-6 demonstrate that variances are not that equal, which also makes me question the presented analysis.
If the authors prefer to stick to the parametric statistics, they must thoroughly explain why they are sure that ANOVA and the following tests can be performed with these data (and maybe also introduce some correction for heteroscedasticity). However, I would encourage the authors to switch to non-parametric statistics and use something like Kruskal–Wallis test (also called one-way ANOVA on ranks in some software) followed by Dunn’s test (or simply by Mann–Whitney tests with Holm’s correction for multiple comparisons) for pair-wise comparisons. In the letter case some differences between groups maybe not statistically significant anymore and it will influence the whole Results and Discussion sections, but overall picture should be stable regardless of the used analysis once it’s performed correctly.
2) Overall, English and formatting are fine and readable, but there are numerous typos, grammatical errors and discrepancies throughout the text. I would like to ask the authors to carefully check every aspect of the manuscript and be consistent. Some specific examples:
- Authors sometimes call Procambarus clarkia a "shrimp". I’m not a native speaker, but "crayfish" seems more relevant. Would you please check it with a qualified translator.
- On panels (a) in figures 3, 4 and 6 gill and hepatopancreas are exchanged. Even if it’s not a mistake, the order should be consistent.
- Lines 24-25 "The activities of catalase (CAT), superoxide dismutase (SOD), malondialdehyde (MDA)". Obviously, MDA has no activity, the term "content" or similar should be used. Please check such inconsistencies throughout the whole text.
- Would you please especially carefully check spelling for all units. For example, hours (h) are sometimes written without space after the number; in line "45*60*30CM" centimeters are probably "cm".
- The term ROS is introduced but not explained. Please spell every term before using the abbreviation.
- Spaces and dots are often missed or excessive.
3) The authors have to be a bit more specific in describing the study and its results. In particular:
- Lines 81-82, the authors talk about the immune response, but what relation does it have to the manuscript? Again, on lines 239-241 HSPs are somehow related to immune response (and they indeed can be related indirectly), but what do you mean by the term «immune» throughout the manuscript?
- Line 114, what enzymes do you mean in the phrase «immunity enzyme parameters»? Later on line 173 only «Antioxidant and metabolic enzyme changes» are mentioned.
- In line 88 the authors state in the goal «preliminary studies on the recovery state of crayfish after air exposure» (grammatically incorrect as far as I can see) and probably mean that this study pioneers the topic. However, I see at least two recent papers from different groups with similar analyses on the same species: https://doi.org/10.1016/j.aaf.2022.01.001 and especially https://doi.org/10.1016/j.dci.2022.104480. I very much recognize that those studies did not include re-submersion after air exposure, but to my opinion they have to be at least mentioned in Introduction. Ideally, the obtained results should be fully compared with data from those works.
- Line 106, authors claim random sampling of animals from the tanks. However, the tanks seem to be relatively small, while the amount of animals (60) seems to be high. Were some animals located on top of others? Can you imagine non-random selection due to this issue?
- Line 304 «This indicates that the antioxidant enzyme SOD was consumed in a large amount during the submersion stage». What do you mean by «consumed» here?
- Line 416 «dry transport … can cause low survival rate». However, the data about mortality during the experiment are not presented at all.
Additional concerns:
- The non-normally distributed data (but normally distributed can be as well) are usually presented as boxplots or as medians (for example, represented by bars) with individual data points. It would be nice if authors consider switching to such a way for data presentation, but this is just a suggestion.
- It was very difficult for me to follow the changes in certain parameters since the air exposure (probably should be labeled on OX axis «Time during air exposure») and re-submersion data are presented on separate figures. I would suggest to re-organize the data and separate them by parameters, not by exposure. For example, CAT+SOD can be presented on one figure, and MDA+LDH on another one.
- It took some time for me to understand that the letters indicating statistically significant differences on Figs 3-6 are separately considered for each tissue. I see that the necessary explanation is added to Figs 3-4 (but not 5-6), yet I would encourage the authors to use separate ranges of letters for different tissues. For example, small Greek letters or other range of English alphabet (possibly starting from k) for gill. This is not mandatory, of course, but it may make reading the figures more straightforward. However, the relevant explanation certainly should be added to the section 2.5 describing the statistical analysis.
- The labels on figures 1-2 are barely readable.
- Line 103, can you provide a reference for the air exposure period?
- Line 110, how the animals were euthanized?
- Lines 72-74, it’s a bit strange to say that crustaceans developed antioxidant defense systems, such systems are known even for bacteria.
- Lines 74-76, similarly, it’s strange to say «Recent studies have shown that many antioxidant defense systems in crustaceans … are essential to maintaining ROS balance». As far as I know, antioxidant enzymes are known in crustaceans for decades (so, not recent), and they far not specific to crustaceans.
Author Response
To REVIEWER
- Point 1: The concern about the current version of manuscript is statistical analysis for the biochemical and expression measurements.
Response 1: Thank you very much for your query and valuable suggestions. We have deeply reflected on your advice, and we agree with you that 12 samples per group does not guarantee the stability of the normality test to some extent. We have reviewed much of the literature and found that there are a number of experiments that do not satisfy the 20 samples(DOI: 10.1016/j.aqrep.2021.100898; DOI: 10.1016/j.cbi.2018.06.012), but only if all of them are tested for normality prior to analysis. We have reprocessed as you advise and have added data from the ignored technical replicates (although the technical replicates are not as convincing as the biological replicates) to minimize the differences between groups. We performed both a nonparametric test using the Kruskal-Wallis test and a Dunn test to ensure compliance with the normal distribution. From the results, the overall trend of enzyme activity and gene changes after air exposure was more consistent with previous studies (DOI: 10.1016/j.dci.2022.104480; DOI: 10.1016/j.aqrep.2021.100898), in which case we believe that our results can be analyzed further.
- Point 2: Authors sometimes call Procambarus clarkia a "shrimp". I’m not a native speaker, but "crayfish" seems more relevant. Would you please check it with a qualified translator.
Response 2: Sorry for the problem. shrimp really doesn't fit description Procambarus clarkia and has been corrected in the article.
- Point 3: On panels (a) in figures 3, 4 and 6 gill and hepatopancreas are exchanged. Even if it’s not a mistake, the order should be consistent.
Response 3: Sorry for the problem. We have recreated all the figures.
- Point 4: Lines 24-25 "The activities of catalase (CAT), superoxide dismutase (SOD), malondialdehyde (MDA)". Obviously, MDA has no activity, the term "content" or similar should be used. Please check such inconsistencies throughout the whole text.
Response 4: Thank you for your reminder. It has been changed to the content of MDA.
- Point 5: Would you please especially carefully check spelling for all units. For example, hours (h) are sometimes written without space after the number; in line "45*60*30CM" centimeters are probably "cm".
Response 5: Thank you for your reminder. The article has been checked and changes have been made.
- Point 6: The term ROS is introduced but not explained. Please spell every term before using the abbreviation.
Response 6: Thank you for the reminder. In the introduction section, an explanation of ROS has been added.
- Point 7: Spaces and dots are often missed or excessive.
Response 7: Thank you for your reminder. The article has been checked and changes have been made.
- Point 8: Lines 81-82, the authors talk about the immune response, but what relation does it have to the manuscript? Again, on lines 239-241 HSPs are somehow related to immune response (and they indeed can be related indirectly), but what do you mean by the term «immune» throughout the manuscript?
Response 8: Thank you for the reminder. Due to our oversight, immunity was not explored in depth in this experiment. Although hsp70 has an immune-related function, the effect of air exposure on immune function was not explored in depth in the experiment. Immunity is a good research direction that we will explore in depth in the subsequent study. To avoid misunderstanding, the content about immunity will be removed from the article.
- Point 9: Line 114, what enzymes do you mean in the phrase «immunity enzyme parameters»? Later on line 173 only «Antioxidant and metabolic enzyme changes» are mentioned.
Response 9: We are sorry for this. The title has been changed.
- Point 10: In line 88 the authors state in the goal «preliminary studies on the recovery state of crayfish after air exposure» (grammatically incorrect as far as I can see) and probably mean that this study pioneers the topic. However, I see at least two recent papers from different groups with similar analyses on the same species: https://doi.org/10.1016/j.aaf.2022.01.001 and especially https://doi.org/10.1016/j.dci.2022.104480. I very much recognize that those studies did not include re-submersion after air exposure, but to my opinion they have to be at least mentioned in Introduction. Ideally, the obtained results should be fully compared with data from those works.
Response 10: Thank you for your suggestions. Our primary objective is to investigate the effect of re-submersion on the recovery of crayfish after air exposure. The two articles you mentioned are indeed of high reference value. We will add a comparison with them in our discussion. e.g., both SOD and CAT showed similar trends in the hepatopancreas under air exposure.
- Point 11: Line 106, authors claim random sampling of animals from the tanks. However, the tanks seem to be relatively small, while the amount of animals (60) seems to be high. Were some animals located on top of others? Can you imagine non-random selection due to this issue?
Response 11: Thank you for your question. The space is in fact sufficient, in fact, during the air exposure phase, the crayfish will hold each other to avoid water loss, which cannot be avoided. We tried to ensure random sampling of each group as much as possible.
- Point 12: Line 304 «This indicates that the antioxidant enzyme SOD was consumed in a large amount during the submersion stage». What do you mean by «consumed» here?
Response 12: Thank you for your question. CONSUMED describes the slow recovery state of SOD, which is caused by a lack of protein oxidation products due to a decrease in metabolic rate.
- Point 13: Line 416 «dry transport … can cause low survival rate». However, the data about mortality during the experiment are not presented at all.
Response 13: Thank you for your question, and strongly agree with you that ‘Low survival’ does not apply here. The survival rate, in this experiment was 100% at both stages. In this experiment, the delivery of crayfish was simulated in the autumn and winter seasons, and the temperature was relatively low, so the mortality in the present experiment was 0. However, according to the combined histological and physiological indicators of the results of this experiment, a certain mortality rate may occur with the extension of the exposure time. However, the results of this article also confirm that the negative effects of crayfish can be improved by the re-submersion operation, thereby improving the health status of crayfish.
Additional concerns:
- Point 14: The non-normally distributed data (but normally distributed can be as well) are usually presented as boxplots or as medians (for example, represented by bars) with individual data points. It would be nice if authors consider switching to such a way for data presentation, but this is just a suggestion.
Response 14: Thank you very much for your suggestion. We think a bar chart would better show the changes in these indicators with time.
- Point 15: It was very difficult for me to follow the changes in certain parameters since the air exposure (probably should be labeled on OX axis «Time during air exposure») and re-submersion data are presented on separate figures. I would suggest to re-organize the data and separate them by parameters, not by exposure. For example, CAT+SOD can be presented on one figure, and MDA+LDH on another one.
Response 15: Thank you for your suggestion, and agree with you very much. We will recreate the picture according to the hepatopancreatic and gill classification.
- Point 16: It took some time for me to understand that the letters indicating statistically significant differences on Figs 3-6 are separately considered for each tissue. I see that the necessary explanation is added to Figs 3-4 (but not 5-6), yet I would encourage the authors to use separate ranges of letters for different tissues. For example, small Greek letters or other range of English alphabet (possibly starting from k) for gill. This is not mandatory, of course, but it may make reading the figures more straightforward. However, the relevant explanation certainly should be added to the section 2.5 describing the statistical analysis.
Response 16: Thank you for your suggestion and sorry for the confusion. We will add the appropriate explanation in section 2.6.
- Point 17: The labels on figures 1-2 are barely readable.
Response 17: Sorry about this, we have zoomed in on the picture letters.
- Point 18: Line 103, can you provide a reference for the air exposure period?
Response 18: Thank you for your query. The production area of crayfish in China is concentrated in the central region, e.g., Jiangsu Province is transported to the breeding area in the south, adding the pre-fishing and packing time, probably the actual time spent within 24 h, so this experiment chose 24 h as the exposure time. And some studies have shown that under room temperature conditions, crayfish die from air exposure for 24 h, and the mortality rate can exceed 15%https://doi.org/10.1016/j.aqrep.2021.100898.
- Point 19: Line 110, how the animals were euthanized?
Response 19: Before sampling, put the crayfish into ice water for a short period of anesthesia until motionless, and start sampling.
- Point 20: Lines 72-74, it’s a bit strange to say that crustaceans developed antioxidant defense systems, such systems are known even for bacteria.
Response 20: Thank you for the reminder. That's not really the right description, it should be the crustaceans have antioxidant system.
- Point 21: Lines 74-76, similarly, it’s strange to say «Recent studies have shown that many antioxidant defense systems in crustaceans … are essential to maintaining ROS balance». As far as I know, antioxidant enzymes are known in crustaceans for decades (so, not recent), and they far not specific to crustaceans.
Response 21: Thank you for your suggestion and I agree with you. There is a great deal of research on crustacean antioxidant systems, and the concept I described is really not a recent one.

Round 2
Reviewer 1 Report
General comments
The manuscript has been revised well. The results are easy to understand because of table addition and figures become clear. The conclusion based on a result is also considered to be a valid. I think this manuscript will be acceptable after some corrections have been done.
Specific comments
Line number 162 table should be "Table".
Line number 206 Fig. F should be "Fig. 4F".
Line number 225 Fig. should be "Figs.".
Line number 199, 200, 202, 229, 231, 233 Since reader can understand hepatopancreas section in these paragraphs, I suggest delete "in the hepatopancreas".
Line number 206, 207, 208, 238, 242, 245 Since reader can understand gills section in these paragraphs, I suggest delete "in the gills".
Figures 3, 4, 5 and 6 I recommend it be modified: different exposure and re-submersion times
Figures 6 Letter at R-3H should be "b".
Table 2 Explanation for each abbreviation (L, B, R, TV, Rec, H and GM).
Author Response
To REVIEWER
Specific comments
- Line number 162 table should be "Table".
Response: Thanks for the reminder, "table" has been changed to "Table" in Line 162.
- Line number 206 Fig. F should be "Fig. 4F".
Response: Thanks for the reminder, “Fig. F” has been changed to “Fig. 4F” in Line 206.
- Line number 225 Fig. should be "Figs.".
Response: Response: Thanks for the reminder, we complete this as Fig. 5 and Fig. 6.
- Line number 199, 200, 202, 229, 231, 233 Since reader can understand hepatopancreas section in these paragraphs, I suggest delete "in the hepatopancreas".
Response: Thank you for your suggestion, we have removed the duplicate descriptions for easier reading.
- Line number 206, 207, 208, 238, 242, 245 Since reader can understand gills section in these paragraphs, I suggest delete "in the gills".
Response: Thank you for your suggestion, we have removed the duplicate descriptions for easier reading.
- Figures 3, 4, 5 and 6 I recommend it be modified: different exposure and re-submersion times.
Response: Thank you for your suggestion. In Fig3,4,5,6, we used 0 h - 24 h to denote the air exposure time and R-1 h - R-24 h to denote the re-submersion phase.
- Figures 6 Letter at R-3H should be "b".
Response: Thank you for the reminder. In Fig.6 E, the flag for R-3 h has been changed to “b”
- Table 2 Explanation for each abbreviation (L, B, R, TV, Rec, H and GM).
Response: Thank you for the reminder, we made a note in the Table 2 that L. Lumen; B. B cell; R. R cell; TV. Transferred vacuoles. Rec. Respiratory epithelium cells; H. Haemocytes; GM. Gill membranes.